# Multi-Branch Gated Fusion Network: A Method That Provides Higher-Quality Images for the USV Perception System in Maritime Hazy Condition

**Yunsheng Fan** [1,2] , **Longhui Niu** [1,2] **and Ting Liu** [1,2,*]

1   School of Marine Electrical Engineering, Dalian Maritime University, Dalian 116026, China
2   Key Laboratory of Technology and System for Intelligent Ships of Liaoning Province, 1 Linghai Road, Dalian 116026, China
*   Correspondence: liuting0910@dlmu.edu.cn; Tel.: +86-155-2478-9899

**Abstract:** Image data acquired by unmanned surface vehicle (USV) perception systems in hazy situations is characterized by low resolution and low contrast, which can seriously affect subsequent high-level vision tasks. To obtain high-definition images under maritime hazy conditions, an end-to-end multi-branch gated fusion network (MGFNet) is proposed. Firstly, residual channel attention, residual pixel attention, and residual spatial attention modules are applied in different branch networks. These attention modules are used to focus on high-frequency image details, thick haze area information, and contrast enhancement, respectively. In addition, the gated fusion subnetworks are proposed to output the importance weight map corresponding to each branch, and the feature maps of three different branches are linearly fused with the importance weight map to help obtain the haze-free image. Then, the network structure is evaluated based on the comparison with pertinent state-of-the-art methods using artificial and actual datasets. The experimental results demonstrate that the proposed network is superior to other previous state-of-the-art methods in the PSNR and SSIM and has a better visual effect in qualitative image comparison. Finally, the network is further applied to the hazy sea–skyline detection task, and advanced results are still achieved.

**Keywords:** maritime image dehazing; multi-branch network; residual attention; gated fusion sub-network; sea–skyline detection

## 1. Introduction

In the past few years, USVs have become widely used in civil domains including maritime search and rescue, in addition to playing a significant role in the sphere of national defense [1–4]. However, the sea navigation conditions are complex, and the ocean images in hazy weather are characterized by low contrast, few high-frequency components, and a large number of missing details [5–7]. This will hinder subsequent advanced visual tasks. Therefore, it is very important for the USV perception system to obtain high-definition images in marine haze areas [8–10].

The reduction in picture quality brought on by maritime haze can be roughly expressed by mathematical Equation (1) [11,12].

$$I(x) = J(x)t(x) + A(x)(1 - t(x)) \tag{1}$$

where $I(x)$ represents the hazy image acquired by the imaging device. $J(x)$ represents the haze-free image. $A$ represents the atmospheric light value. $t(x) = e^{-\rho d(x)}$ represents the transmission map [13]. The inverse restoration procedure of the physical deterioration process described in Formula (1) is called maritime image dehazing, which is a highly ill-posed problem because of the unknown transmission map and global atmospheric light. To address this difficult issue, a wide variety of image dehazing methods have been put

out in recent years. These methods are broadly classified into traditional a priori-based methods and modern deep-learning-based methods. The handcrafted nature of the picture priors in the former type is the primary distinction between the two types, while the image priors of the latter type are learned automatically.

Traditional dehazing methods mainly use a priori knowledge to dehaze. For example, He et al. [14] proposed the dark channel prior (DCP). He et al. believe that there is always a channel with a very low gray value in the hazy map. Based on this prior knowledge, the transmission map is solved, and then, the atmospheric scattering model is used for image dehazing. This method achieved a good effect at that time, but it will cause color distortion in some scenes. Meng et al. [15] proposed an effective regularization method to remove haze. This dehazing method based on a boundary constraint can solve the problem of the low brightness of an image. Zhu et al. [16] proposed an a priori method of the color attenuation prior (CAP). The above early image dehazing methods can achieve good results in some specific scenes and have made a great contribution to the development of dehazing technology. However, because most of these methods rely on a priori information, the accuracy of the accepted assumptions/priors with regard to the target scenarios naturally limits their performances. Therefore, the traditional method cannot achieve the expected effect in many cases.

Deep learning techniques attempt to directly regress the final haze-free image or the intermediate transmission map; this overcomes the limitation of a specific prior. With the Big Data being applied, using their strong representation and end-to-end learning ability, people have proposed many image dehazing methods based on deep convolution neural networks (CNNs) [17–26] and achieved superior performance and robustness. For example, DehazeNet, an end-to-end trainable deep CNN model, was proposed by Cai et al. [27], which can learn and transmit directly from a hazy image, which is better than modern a priori methods and the random forest model. A multiscale CNN (MSCNN) was suggested by Ren et al. [17] to learn transmission maps in the way of full convolution and explored a multiscale structure for coarse-stage to fine-stage regression. The dense connected pyramidal dehazing network (DCPDN) was proposed by Zhang et al. [18] to concurrently learn the transmission map and atmospheric light. The method also utilizes adversarial loss based on generative adversarial networks [19] to supervise the dehazing network. One problem with these CNN-based methods is that all of them require accurate transmission maps and atmospheric light. Li et al. [20] did not estimate an intermediate transmission map. An end-to-end CNN model, called the all-in-one dehazing network (AOD-Net), was proposed for learning haze-free images from hazy images. Despite the reformulation of the haze imaging model by integrating transmitted and atmospheric light into a single variable, there is still a need to accurately estimate the intermediate variables, so that AOD-Net still belongs to the physical model in (1). Learning the image a priori through the deep learning method has largely gotten rid of the limitations of traditional methods, but it still follows the traditional dehazing model. Therefore, if we cannot accurately estimate the image a priori, it will still lead to low-quality results.

Different from the CNN method for estimating intermediate variables, the network proposed in [21–25] is constructed based on the principle of image fusion. Instead of estimating the intermediate parameters of the atmospheric scattering model, it learns the relevant features of the image and feature fusion through the network, then directly restores the haze-free image. In general, fusing features from different levels can improve network performance. To implement this idea, Reference [21] used feature pyramids to combine low-level and high-level semantic feature maps to carry out this approach, and Reference [18] used dense connected pyramidal networks to achieve feature fusion at all scales. Hang Dong [22] proposed MSBDN-DFF based on the U-Net architecture with dense feature fusion. The network integrates the dense feature fusion module (DFF) into the U-Net architecture to make up for the missing spatial information in high-resolution features at the same time. Ayush [23] proposed a novel generative countermeasure network structure: the back-projected pyramid network (BPPNet). The spatial context is reserved

through the iterative block of U-Net, and the multi-scale structure information is fused through a new pyramid convolution block. Most of these feature fusion methods are implemented by brutally overlaying pixels, treating each level of the feature map without any difference and easily losing important feature information. Therefore, Ren et al. [24] proposed a gated fusion network (GFN) using an encoder–decoder architecture. Learning three pre-processed images obtained from the original image, the proportion of these three image features in the output image is automatically obtained using the gated fusion network and fused to recover the haze-free image. Tan et al. [25] proposed a multi-branch deep fusion network (MBDF-Net) for 3D target detection, and a simple adaptive attention fusion (AAF) module was designed in the feature extraction phase to suppress non-interest regions and feature fusion. Considering the real-time nature of the algorithm, Shi et al. [26] proposed an efficient and concise multi-branch fusion convolution network for remote sensing image scene classification. In the network, SE and CBAM attention modules are added to extract shallow and deep features, respectively, and fusion is performed according to the generated attention weight. A gated contextual aggregation network (GCANet) was suggested by Chen et al. [28] to directly reconstruct the final haze-free image. The method proposes a more lightweight and brief gated fusion sub-network to fuse characteristics at several levels, assigning importance weights to the elements at varying levels. When compared to conventional approaches, these deep-learning-based methods circumvent the issue of low image restoration quality caused by image prior estimation error and improve the flexibility of feature image fusion. However, the lack of the differentiated treatment of channel, pixel, and spatial features in feature extraction hinders the characterization capability of deep networks.

To address the above problems, based on the principle of image fusion and the characteristics of marine hazy image, this paper proposes a new end-to-end multi-branch feature gated fusion network, which is used for maritime image dehazing and directly obtains haze-free images. The network proposes three residual attention modules each incorporating different branch networks to concentrate on key elements and optimize the details of the image. To determine the relative importance of various branch feature maps, the weight maps of the corresponding branches are adaptively learned using the gated fusion sub-network and to combine the feature maps using the associated importance weights. Compared with other dehazing methods, it performs outstandingly in preserving image colors and recovering information in thick haze areas. The experiments show that MGFNet has better performance than previous image dehazing methods, both qualitatively and quantitatively. A comprehensive ablation study is also provided in the paper to verify the significance and requirement of each element. Further, the proposed MGFNet is applied to the hazy sea–skyline detection task, which performs superior to the previous state-of-the-art network.

The following are the paper's contributions:

① Three different residual attention modules are proposed, which are applied to different branch networks based on U-Net [29], and three inputs are extracted from the original image through the branch network: The first input is obtained by incorporating the branch network of residual channel attention, which aims to weaken the color distortion caused by atmospheric light scattering. The second input is obtained by fusing the branch network of residual spatial attention to enhance the contrast and produce better global visibility. In order to recover the detail of thick haze region, the third input fuses the residual pixel attention and focuses on the thick haze pixel region.

② The gated fusion sub-network is proposed to adaptively learn the weight maps of corresponding branches to determine the importance of different branch feature maps and to fuse the feature maps according to their corresponding importance weights. The purpose is to seamlessly fuse the branch feature graph by retaining specific features and enhance the network performance.

The structure of this paper is as follows: Section 1 introduces the dehazing model and discusses pervious traditional and modern deep-learning-based dehazing methods;

Section 2 discusses the proposed multi-branch gated fusion network and the components of the network; Section 3 introduces the loss function for the training network; Section 4 compares and analyzes the results of the comparative experiments; Section 5 applies the proposed method to sea–skyline detection to further verify the advanced nature of the method. Section 6 compares and analyzes average model runtime. Section 7 concludes this paper.

## 2. Proposed Method

The architecture of the proposed multi-branch gated fusion network (MGFNet) is shown in Figure 1. The input of the MGFNet is a single hazy image, and the output is a recovered haze-free image. MGFNet is composed of a shallow feature extraction module, a multi-branch feature extraction network, and a feature fusion module, which uses a ReLU activation function and two 3 × 3 convolution layers to extract the fundamental elements of the image; the input image channel is 3, and the feature image channel is expanded to 96 channels through the shallow feature extraction module. For the purpose of extracting key features and feature map reconstruction, the feature map is input separately into the branch network. Subsequently, the output features of the multi-branch network are passed through a gated fusion subnet for branch information integration. Finally, the fused feature maps are aggregated through a 1 × 1 convolutional layer to recover the images.

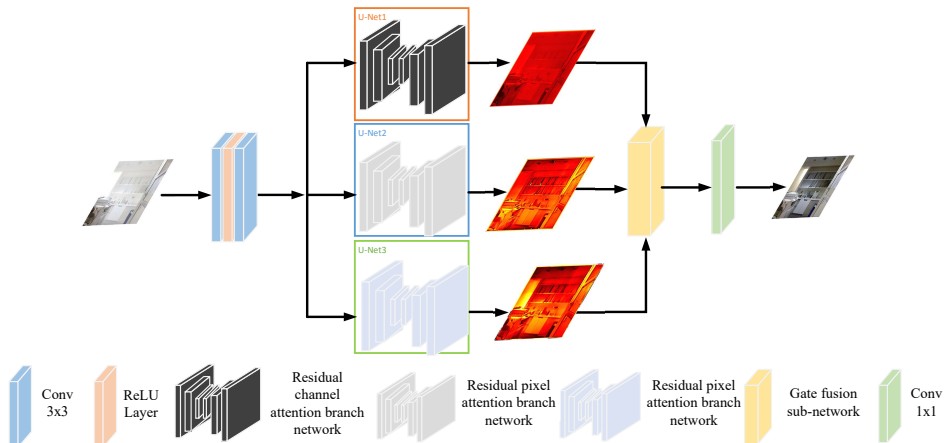

**Figure 1.** Multi-branch gated fusion network architecture.

### 2.1. Residual Attention Branch Network

The multi-branch network consists of three residual attention branch networks similar to U-Net. The input is a shallow feature map extracted by two convolutions and a ReLU activation function. All branch networks have a three-tier U-Net architecture. The main difference between these three branch networks is the type of residual attention blocks used in the branch network. Using the three existing types of attention blocks, it is proposed to include residual pixel attention blocks, residual channel attention blocks, and residual spatial attention blocks. Different residual attention blocks are used to replace the ordinary convolution in each feature branch, so that the network can focus more on the area with dense haze and crucial information, so as to make the image feature expression more distinct. The whole encoder–decoder branch network contains 7 residual attention blocks, 3 skip connection structures, 2 downsampling layers, and 2 upsampling layers. Taking the residual channel attention branch network model as an example, the design details are shown in Table 1.

**Table 1.** Detailed model design of the residual attention branch network.

| Module | Module Details | Input Shape | Output Shape |
|---|---|---|---|
| Residual Attention Block1 | Conv3x3, ReLU, Conv3x3, CA Layer | 96, W, H | 96, W, H |
| Skip Connection1 | Conv3x3, ReLU, Conv3x3, CA Layer | 96, W, H | 96, W, H |
| Downsample1 | $Upsample(scale_factor = 1/2)$, Conv1x1 | 96, W, H | 144, W/2, H/2 |
| Residual Attention Block2 | Conv3x3, ReLU, Conv3x3, CA Layer | 144, W/2, H/2 | 144, W/2, H/2 |
| Skip Connection2 | Conv3x3, ReLU, Conv3x3, CA Layer | 144, W/2, H/2 | 144, W/2, H/2 |
| Downsample2 | $Upsample(scale_factor = 1/2)$, Conv1x1 | 144, W/2, H/2 | 192, W/4, H/4 |
| Skip Connection3 | Conv3x3, ReLU, Conv3x3, CA Layer | 192, W/4, H/4 | 192, W/4, H/4 |
| Upsample2 | $Upsample(scale_factor = 2)$, Conv1x1 | 192, W/4, H/4 | 144, W/2, H/2 |
| Concat2 | Concat(Skip Connection2, Up Sample1) | 144, W/2, H/2 | 288, W/2, H/2 |
| Residual Attention Block3 | Conv3x3, ReLU, Conv3x3, CA Layer | 288, W/2, H/2 | 144, W/2, H/2 |
| Upsample1 | $Upsample(scale_factor = 2)$, Conv1x1 | 144, W/2, H/2 | 96, W, H |
| Concat1 | Concat(Skip Connection1, Up Sample2) | 96, W, H | 192, W, H |
| Residual Attention Block4 | Conv3x3, ReLU, Conv3x3, CA Layer | 192, W, H | 96, W, H |

The encoder part includes 2 residual attention blocks and 2 downsampling layers. The residual attention block consists of a residual structure and an attention module, where the residual structure consists of a convolutional layer with a convolutional kernel of $3 \times 3$ and a ReLU function composition. Then, the input enters the attention module through a $3 \times 3$ convolution. The number of encoder output channels is 96, 144, and 192. In general, the encoder module encodes the image information in a pyramidal scale to obtain multiple scales of feature maps.

The decoder part also contains 2 convolutional layers and 2 upsampling layers. The residual attention module is still used instead of the convolutional layer, and then, the upsampling operation is performed, which includes the upsampling layer with a sampling kernel of $2 \times 2$ and a convolution layer of $1 \times 1$. In contrast to the encoder, the decoder's upsampling layer sequentially recovers the image details, and the final output is the same size as the input image, with the number of output channels 192, 144, and 96, respectively. Meanwhile, to maximize the flow of information between multiple levels, the feature maps between encoders and decoders are connected using skip connections. The residual attention module is added along with the skip connection, which makes the important feature information in the images of each layer fuse and improves the network efficiency.

The residual attention framework consists of a residual structure and an attention module, and the residual connections can bypass unimportant information such as thin haze or low-frequency regions and focus on thick haze regions and high-frequency color information. The residual attention framework is shown in Figure 2. Subsequent ablation experiments demonstrate how its structure might enhance network efficiency and training stability.

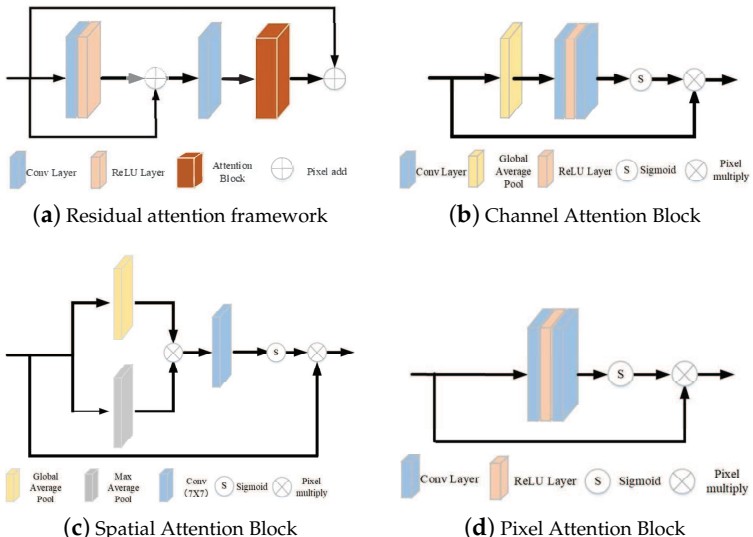

**Figure 2.** Residual attention framework.

A. Channel attention:

The channel attention block is added in the residual framework for the purpose of mitigating the image distortion caused by atmospheric light scattering. The structure is shown in Figure 2b. The channel attention uses global average pooling (GAP) to compress the input features from 3D to 1D, generating a 1D attention mask ($M \in R^C$). Due to the existence of GAP, the output features have global information and are mainly concerned with the color distribution and edge information of the image. Firstly, we created channel descriptors from the channel-related global spatial information using global average pooling, as shown by Equation (2).

$$g_c = H_p(F_c) = \frac{1}{H \times W} \sum_{i=1}^{H} \sum_{j=1}^{W} X_c(i,j) \tag{2}$$

where $X_c(i,j)$ represents the value at the pixel point $(i,j)$ in the feature map of the $C - th$ channel and $H_p$ is the global pooling function. The feature map transforms from $C \times H \times W$ to $C \times 1 \times 1$. The features are passed through two convolutional layers and a sigmoid and a ReLU activation function in order to determine the weights of the various channels, as shown in Equation (3).

$$CA_c = \sigma(Conv(\delta(Conv(g_c)))) \tag{3}$$

where the ReLU function is represented by $\delta$ and the sigmoid function by $\sigma$. The weights of channel $CA_c$ and input $F_c$ are finally multiplied by elements, as shown in Equation (4).

$$F_C^* = CA_c \otimes F_c \tag{4}$$

B. Spatial attention:

Spatial attention blocks are applied in the residual framework with the aim of focusing on image spatial information and improving contrast. Spatial attention can generate two-dimensional attention masks ($M \in R^{H \times W}$). The structure is shown in Figure 2c. Similar to channel notation, it also consists of a global average pooling (GAP) operation or a maximum average pooling (MAP) operation that compresses the features $F_s$ into two dimensions to obtain two $1 \times H \times W$ channel description and stitches them together by channel. The weight coefficients $SA$ are then produced using $7 \times 7$ convolution layers and the sigmoid activation function. Due to the existence of the pooling operation, the output characteristics have non-local information, which mostly concentrates on the image's global

information, such as the change of image brightness. The specific operation is shown in Equation (5).

$$SA = \sigma\Big(Conv^{7x7}([AvgPool(F_s), MaxPool(F_s)])\Big) \tag{5}$$

Finally, the feature map $F_p^*$ is obtained by multiplying the spatial weight $SA$ and the input feature map $F_s$. To offer $F_p^*$ spatial attention in this procedure, $SA$ and $F_s$ are multiplied at the pixel level, as shown in Equation (6).

$$F_s^* = SA \otimes F_s \tag{6}$$

C. Pixel attention:

The 3D attention mask $(M \in R^{H \times W \times C})$ generated after the pixel attention block does not have any pooling or sampling, which means that the output feature map has local information and focuses on high-frequency, dense haze pixel regions in the image. The structure is shown in Figure 2d. The input feature map is routed through two convolutional layers, a nonlinear ReLU activation function and a sigmoid activation function to determine the weight information for each pixel, as indicated by Equation (7).

$$PA = \sigma\big(Conv(\delta(Conv(F_p)))\big) \tag{7}$$

Finally, the pixel weights $PA$ are multiplied with the input feature map $F_p$ pixel by pixel to obtain the feature map $F_p^*$, as shown in Equation (8).

$$F_p^* = F_p \otimes PA \tag{8}$$

*2.2. Gated Fusion Sub-Network*

In order to further achieve effective fusion of local and global features of images, this paper uses gated fusion sub-networks to adaptively fuse multiple feature maps obtained based on branch networks. The core idea of the gated fusion sub-network is to learn the weights of feature maps by convolutional operations, which can make the network automatically focus on the important feature maps and, thus, obtain rich color and structure information. GFN [25] preprocesses the hazy image into 3 derived images, extracts features through the encoder structure, and automatically obtains the weights of the 3 derived images using a gate fusion network in the decoder process, reducing the negative effects of over-enhanced images; GCANet [29] extracts the semantic features of the bottleneck layer of the automatic encoder structure through smooth hole convolution, and the gated fusion sub-network is used to adaptively combine the characteristic images of different depths to improve the clarity of the hazy image.

The difference is that, in this paper, feature maps are obtained by residual attention branch networks, and these branch feature maps fuse the local and global features of the image. Therefore, these features already contain richer color and structure information. In this paper, a simpler structured gated fusion sub-network is used for weight regression of different branch feature maps, focusing on important feature maps and reducing the loss of structural and texture information. The network structure is shown in Figure 3.

Specifically, feature mappings $(F_1, F_2, F_3)$ are first extracted from the different branches and input into the gated fusion sub-network. The outputs of the gated fusion sub-network are three weight maps $(W_1, W_2, W_3)$ corresponding to three feature levels separately obtained by convolution. Finally, the feature mappings of these three different branches are linearly combined with the output importance weight maps. This was performed as shown in Equations (9) and (10).

$$(W_1, W_2, W_3) = \partial(F_1, F_2, F_3) \tag{9}$$

$$F_0 = W_1 * F_1 + W_2 * F_2 + W_3 * F_3 \tag{10}$$

Then, the synthesized feature mapping $F_0$ is fed into the recovered convolutional layer to obtain the target haze-free image. In the paper, the gated fusion sub-network consists of a

convolutional network with a convolutional kernel of $3 \times 3$, and the input is a cascade of three feature mappings, while the number of output channels is 3.

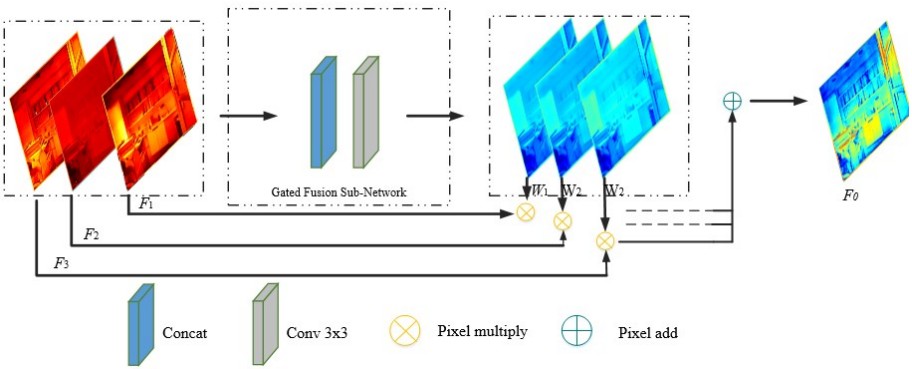

**Figure 3.** Gated fusion sub-network.

### 3. Loss Function

Since the objective evaluation index PSNR now serves as the primary measure of image dehazing, the mean absolute error (MAE) or mean-squared error (MSE) are typically utilized as the loss functions to optimize the network. However, both the L1 and L2 losses are pixel losses that do not take global information into account, so over-smoothing problems are usually encountered. Many loss functions have been proposed, such as perceptual loss [30], adversarial. [19], and even compound loss, which combines multiple loss functions together to solve the problem. In order to focus on the trade-off between human perception and metric scores, new loss functions are proposed to optimize MGFNet, as shown in the following Equation (11).

$$L_{total} = L_{ps}(X, Y) + \alpha L_{edge} \tag{11}$$

where $X \in R^{B \times C \times H \times W}$ represents the degraded image, $B$ represents the batch size of the training data, $C$ is the number of feature channels, and $H$ and $W$ are the image size. $Y$ represents the haze-free image. The total loss function consists of $L_{ps}$ and $L_{edge}$. $L_{ps}$ is the proposed loss consisting of the peak signal-to-noise ratio (PSNR) and structural similarity (SSIM) index, using only two standard metrics as parameters to achieve better visual perception using fewer parameters, as shown in Equation (12).

$$L_{ps} = \frac{1 - SSIM(X, Y)}{PSNR(X, Y) + \omega} \tag{12}$$

where the parameter $\omega$ is a constant empirically set to 0.005. the second term $L_{edge}$ is the edge loss, cited in [31], which limits the difference between the actual image and the anticipated haze-free image's high-frequency component. It can be expressed as Equation (13):

$$L_{edge} = \sqrt{\|\Delta(X) - \Delta(Y)\|^2 + \varepsilon^2} \tag{13}$$

where $\Delta$ denotes the Laplace operator. The constants in Equation (12) and in Equation (13) were set to 0.05.

We also supplemented the comparison with other common loss functions (e.g., MAE loss, MSE loss, MAE and MSE combination loss, adversarial loss) to prove the superiority of the loss function proposed in this paper. The comparison results are shown in Table 2.

It can be seen from Table 2 that different loss functions are used for training under the same parameters and compilation environment. The models PSNR and SSIM trained by the loss function in this paper are higher than other losses, reflecting the superiority of the loss function proposed in this paper.

**Table 2.** Network performance comparison of different loss functions.

| Loss Function | SSIM | PSNR |
| --- | --- | --- |
| MAE loss | 26.7666 | 0.8723 |
| MSE loss | 27.9579 | 0.9047 |
| MSE+MAE loss | 27.8279 | 0.9173 |
| Adversarial loss | 27.5613 | 0.8165 |
| **Ours** | 28.0282 | 0.9278 |

## 4. Experimental Results and Analysis

### 4.1. Experimental Details

The MGFNet was implemented by PyTorch 1.8.0 and an NVIDIA GTX 3090 GPU. As the training model for dehazing, the number of feature channels was set to 96. The optimizer was chosen as Adam, where the default values of 0.9 and 0.999 were used, respectively; this network was trained on a $256 \times 256$ image size; the batch size was configured as four; the number of iteration rounds of the training set was set to 500; the initial learning rate was configured as $2 \times 10^{-4}$; the learning rate can be adjusted using the cosine quenching function [32].

### 4.2. Photoelectric Perceptual System Experimental Platform

This paper took the "Lanxin" USV perception system of Dalian Maritime University as the theoretical subject of research on key perception technologies. The perception system experimental platform is shown in Figure 4. The system consists of an optical camera, LiDAR, marine radar, inertial navigation, thermal imager, and compass. The image obtained by the optical camera contains more abundant target and background information. The target information in the region can be more easily retrieved by the target region segmentation and detection method. LiDAR information may precisely determine the target's three-dimensional position information and pinpoint the target. The latest marine radar uses FMCW and advanced digital signal processing technology to realize obstacle recognition and target tracking by transmitting and receiving electromagnetic wave signals. The thermal imager uses the difference of heat emitted by objects to provide clear images under low-light and no-light conditions. Through these sensors, the environment perception system was built to provide security for the USV's autonomous navigation system.

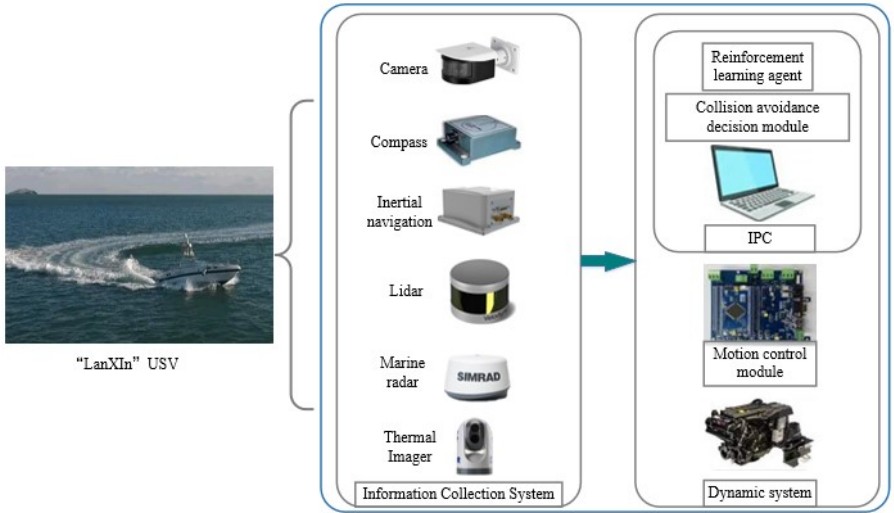

**Figure 4.** "Lanxin" USV perception system experimental platform.

### 4.3. Experimental Dataset

Given that the complex sea state environment in the maritime context is different from that on land and there are additional disturbances in the dehazing process, this paper prepared a dataset as a training set, which contained two parts. Because of the authority of the public dataset, this paper first used part of the training set from the public dataset RESIDE as the first part of the dataset. It included 1339 clear images and 13,990 corresponding hazy images. The hazy images were synthesized from the original images using an atmospheric scattering model, where the global atmospheric light values were taken in the range of 0.8 to 1.0, and the atmospheric scattering parameters were taken in the range of 0.04 to 0.2. However, the problem of interest in this paper is maritime image dehazing, so we prepared another synthetic dataset as the second part of the training set in this paper and jointly trained the network to evaluate the dehazing performance under maritime hazy weather conditions. In the second part of the dataset creation phase, the images after hazy days and their corresponding haze-free images needed to be simulated, and this paper used laboratory UAV photography to obtain clear images of the maritime surface and USV, as shown in Figure 5. An example of an original image and a hazy image is shown in Figure 6. The hazy day effect was well approximated by the atmospheric scattering model (1), which was used to generate hazy images of the maritime surface. Similar to the RESIDE dataset, the range of the global atmospheric light values was 0.8 to 1.0, while the range of the atmospheric scattering parameters was 0.04 to 0.2. In total, the dataset consisted of 500 RESIDE synthetic images and 500 maritime surface synthetic images, of which 800 images served as the training set and 200 images served as the test set.

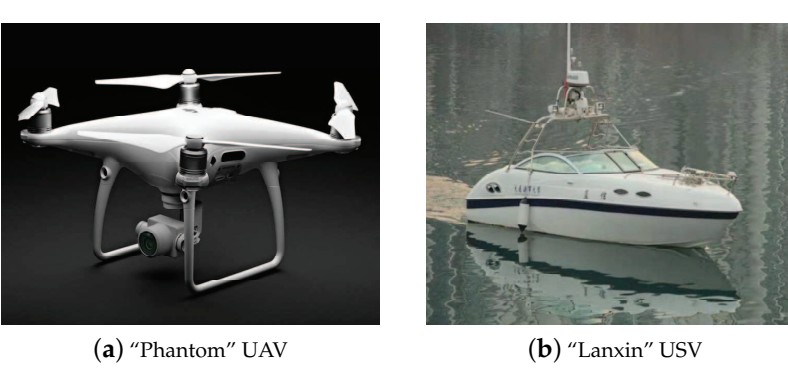

(**a**) "Phantom" UAV                    (**b**) "Lanxin" USV

**Figure 5.** "Phantom" UAV and "Lanxin" USV in the laboratory.

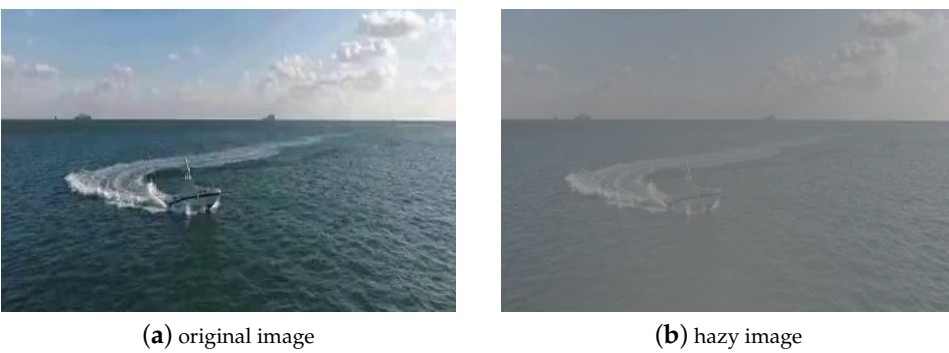

(**a**) original image                    (**b**) hazy image

**Figure 6.** Experimental dataset examples in the laboratory.

### 4.4. Experiment Results

This section compares the dehazing performance of this network with the prior state-of-the-art networks to demonstrate the effectiveness of the network in this paper. The image assessment indices structural similarity (SSIM) and peak signal-to-noise ratio (PSNR) [24]

were introduced for quantitative analysis to intuitively express the dehazing effect of the network. The PSNR is an image quality evaluation index based on the error between corresponding pixels. The units of the PSNR are dB. The larger the value, the smaller the image distortion. It is often simply defined by the mean-squared error (MSE). For example, for two $m * n$ monochromatic images $I$ and $K$, the $MSE$ represents the mean-squared error of the current image $I$ and the reference image $K$, then the $MSE$ is defined as Equation (14):

$$MSE(I, K) = \frac{1}{nm} \sum_{i=0}^{m-1} \sum_{j=0}^{n-1} [I(i,j) - K(i,j)]^2 \tag{14}$$

The $PSNR$ is defined as Equation (15):

$$PSNR(I, K) = 10\log_{10} \frac{255^2}{MSE(I, K)} \tag{15}$$

Structural similarity (SSIM) is another image quality evaluation index, which measures image similarity from brightness, contrast, and structure. When SSIM calculates the difference between two images at each position, it does not take a pixel from each of the two images at that position, but takes a pixel from each region. $SSIM \leq 1$; the larger the $SSIM$, the more similar the two images are. The calculation formula is as Equation (16):

$$SSIM = \frac{(2\mu_x\mu_y + c_1)(\sigma_{xy} + c_2)}{\left(\mu_x^2 + \mu_y^2 + c_1\right)\left(\sigma_x^2 + \sigma_y^2 + c_2\right)} \tag{16}$$

where $\mu_x$ is the average of $x$, $\mu_y$ is the average of $y$, $\sigma_x^2$ is the variance of $x$, $\sigma_y^2$ is the variance of $y$, and $\sigma_x y$ is the covariance of $x$ and $y$. $c_1$ and $c_2$ are constants to maintain stability.

The comparison networks include DCP [14], DehazeNet [27], AOD-Net [20], and GCANet [28], and for a fair comparison, all methods were practiced in the same setting, as described in Section 4.1 for the specific training details. The comparison results are shown in Table 3.

**Table 3.** Quantitative comparison of image dehazing from synthetic datasets.

| Metrics | DCP | DehazeNet | AOD-Net | GCANet | Ours |
|---------|--------|-----------|---------|--------|--------|
| SSIM | 0.8286 | 0.8412 | 0.8367 | 0.8648 | 0.9452 |
| PSNR | 18.93 | 22.31 | 21.14 | 25.47 | 28.33 |

It is evident that the proposed network outperformed other comparative networks in terms of the PSNR and SSIM. In addition, the visual effects of the proposed network were compared with other advanced networks for qualitative comparison. The comparison results are shown in Figure 7.

It is evident that since the DCP follows the dark channel a priori principle, this network cannot accurately estimate the haze thickness when processing hazy images, resulting in the restored images being usually darker than the real-world images (the maritime surface region in the first image and the sea wave light region in the third image in Figure 7b), and there are color distortions in some regions (the sky region in the sixth image in Figure 7b), which eventually affect the visual effect. Since DehazeNet and AOD-Net, two deep-learning-based networks, are not constrained by prior knowledge, the dehazing effect is closer to the actual situation than the DCP network. Nevertheless, the images processed by these two networks still have some residual haze (as shown in Figure 7c,d). The main reason for this phenomenon is the large error in the estimated transmission maps obtained by these two networks in the process of learning the hazy image features, and the overall tone of the image after dehazing is extreme (too bright or too dark). Although GCANet can frequently produce positive results, the recovery ability of the network to the thick haze

region is limited, and there is still undeleted haze in the thick haze region (as shown the yacht region in the second picture of Figure 7e). Moreover, the color retention ability is insufficient in the high-frequency region (as shown by the cloud area in the sixth image of Figure 7e). In contrast, as the proposed network incorporates a residual attention block, it is able to process different channels and pixel information differently, giving more attention to the thick haze region. Therefore, the network can better handle regions with a high haze concentration, while retaining the high-frequency information of the original image clearly and reducing the color distortion.

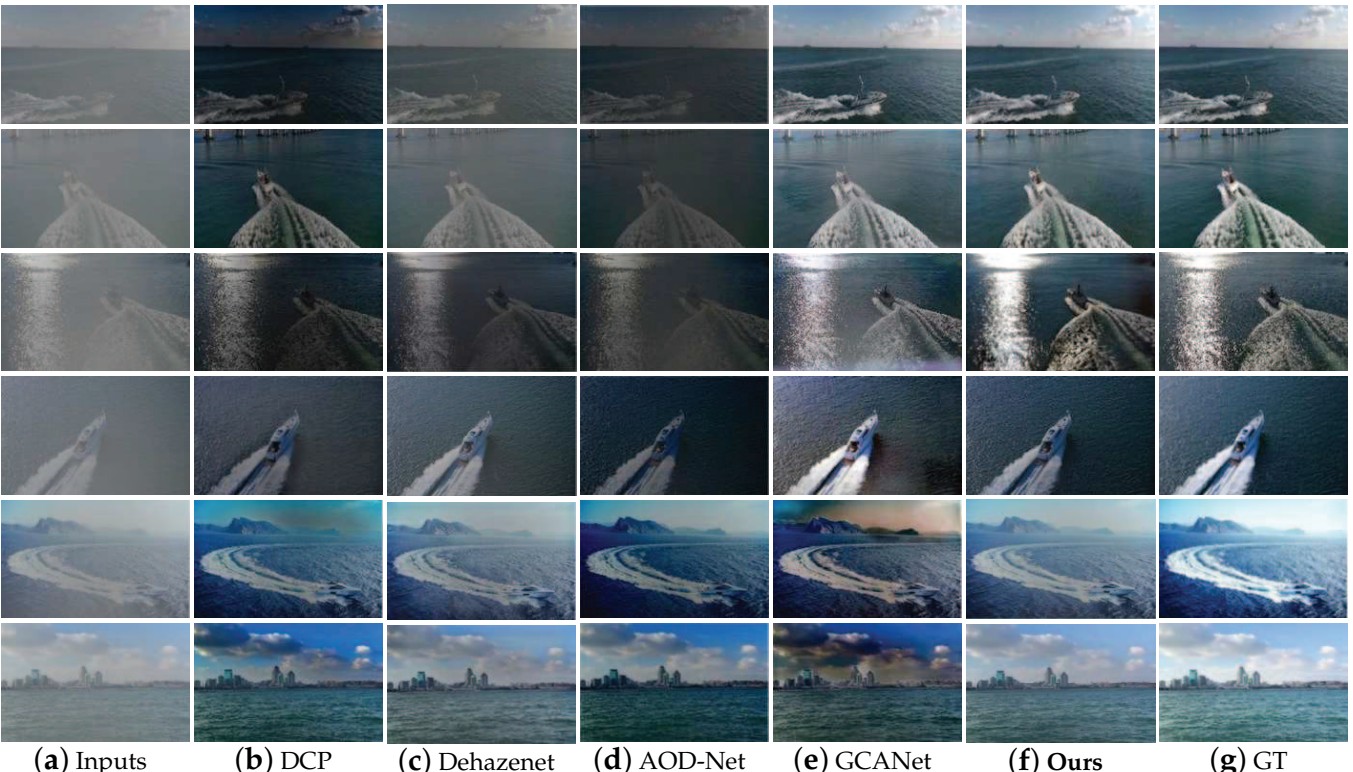

|**(a)** Inputs|**(b)** DCP|**(c)** Dehazenet|**(d)** AOD-Net|**(e)** GCANet|**(f)** **Ours**|**(g)** GT|

**Figure 7.** Qualitative comparison of different dehazing networks in synthetic images (GT: ground truth).

In order to confirm the proposed network's dehazing effect in actual scenes, the proposed network was compared qualitatively with the comparison networks on four real-world images, and the outcomes of the experiment are displayed in Figure 8.

It is evident that the images processed by the DCP still have dull colors and severe color distortion problems. DehazeNet, AOD-Net, and GCANet also show varying degrees of color distortion and even black shadows (As the sky region in the first image of Figure 8e). However, the proposed network results almost exactly match the real scene information, and it performs significantly better than other networks in terms of color retention in high-frequency regions and detail recovery in thick haze regions (as the distant peaks in Figure 8f), while the visual effect is better.

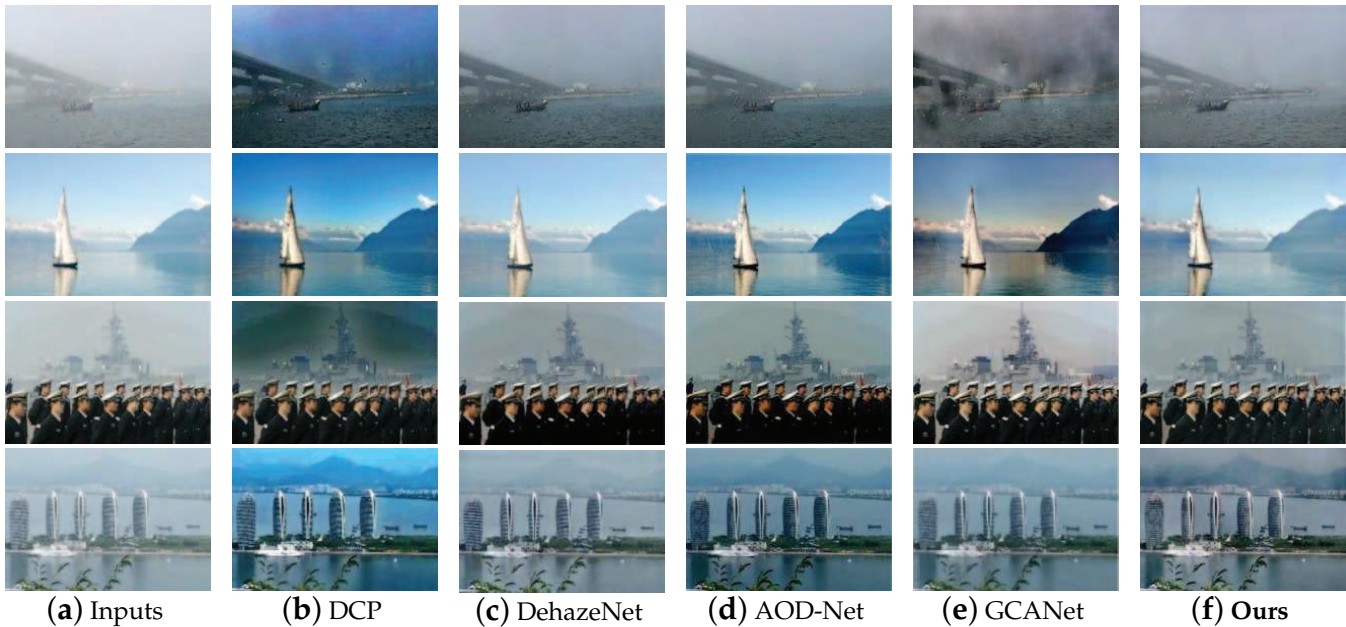

| **(a)** Inputs | **(b)** DCP | **(c)** DehazeNet | **(d)** AOD-Net | **(e)** GCANet | **(f) Ours** |

**Figure 8.** Qualitative comparison of different networks on real-world images.

### 4.5. Ablation Analysis

To comprehend the significance of each MGFNet component, we carried out independent studies of ablation with and without each unique component. We paid particular attention to three key elements: with/without residual structure, with/without attention mechanism, and with/without gated fusion subnets. Accordingly, the image dehazing task was tested on three different network topologies, with each configuration adding one component at a time incrementally. The experimental details were the same as for the setup in Section 4.1. The outcomes of the tests with ablation are displayed in Table 4.

**Table 4.** Ablation analysis for each component with different training configurations.

| Residual structure | ✗ | ✓ | ✓ | ✓ |
|---|---|---|---|---|
| Attention module | ✗ | ✗ | ✓ | ✓ |
| Gate fusion sub-network | ✗ | ✗ | ✗ | ✓ |
| PSNR | 20.87 | 22.36 | 27.44 | 28.16 |

It is clear from these experiments that the final performance of the network improved. In other words, by integrating each of the component parts that were designed, greater gains can be obtained than applying only one or some of them, especially the residual attention module, which can keep the network very competitive.

## 5. Effect Evaluation of Sea–Skyline Detection on Dehazing Network

The sea–skyline is an important factor for USV vision technology to perceive the surrounding environment. Accurate maritime antenna detection can accurately divide the sea–sky region, which is important for the safe navigation of USVs on the maritime surface and target detection [33–36]. Liang et al. [37] proposed a sea–skyline detection method for complex environments. This method locates the sea–sky area based on texture features and uses the OTSU algorithm to obtain an adaptive segmentation threshold to generate a group of sea–skyline candidate points. Finally, a simple clustering method is used to select appropriate points and transform them by line fitting. This method can accurately detect the sea wave line under the complex background of many clouds and waves, but it does not perform well under the maritime haze condition, mainly because the image contrast is low and the texture features are not obvious enough in maritime haze conditions.

In maritime hazy weather, the elimination of haze is an important pre-processing step for sea–skyline detection. In other words, the accuracy of sea–skyline detection is a valid metric to estimate the performance of the dehazing network. The Hough transform [38] or Radon transform [39] are computationally small and have anti-interference. The fundamental needs of the USV can be satisfied by fitting the edge pixels to the sea–skyline using the Hough transform. Therefore, in this paper, fitting the edge pixels made use of the Hough transform, and the image quality was evaluated by counting the accuracy of the sea–skyline detection in the images before and after dehazing, respectively. Figure 9 shows the effect of the sea–skyline detection.

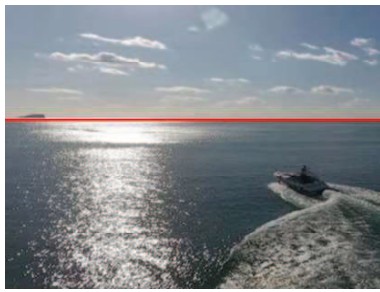 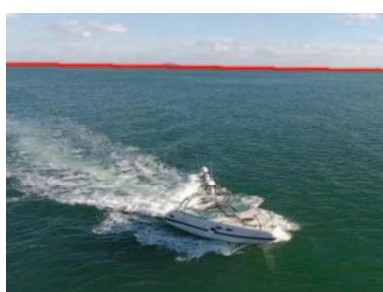

**Figure 9.** The effect of sea–skyline detection.

Statistical calculation and analysis of the sea–skyline detection results for a certain amount of maritime surface images were performed. Judgment basis for the maritime surface image: when the detected sea–skyline point is within five pixels from the real sea antenna, the sea antenna is considered to be correctly detected; otherwise, it is judged to be incorrectly detected. The real sea–skyline is artificially labeled. The quantitative detection accuracy is counted, and the calculation procedure is as follows in Equation (17).

$$p = \frac{q_0}{q_s} \tag{17}$$

where $p$ denotes the accuracy of sea–skyline detection, $q_0$ denotes the number of correctly detected images, and $q_s$ denotes the total number of selected maritime surface images. In order to evaluate whether the proposed network is effective, 20 real maritime surface images with haze were randomly selected, and the above sea–skyline detection algorithm was used to detect the sea–skyline on the images before and after the haze removal. Figure 10 shows the effect comparison of each network after dehazing, and the comparison of the accuracy is shown in Table 5.

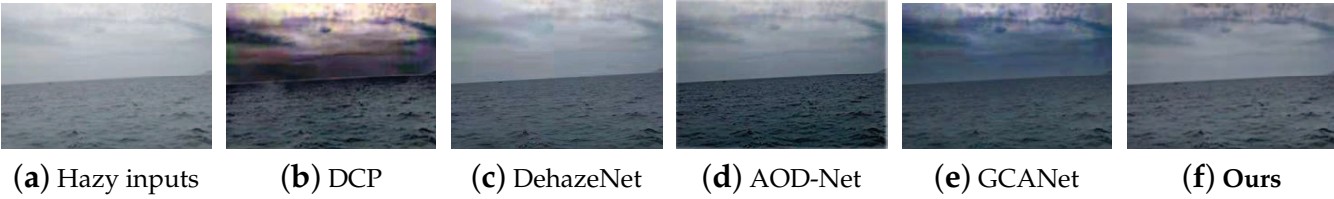

**(a)** Hazy inputs    **(b)** DCP    **(c)** DehazeNet    **(d)** AOD-Net    **(e)** GCANet    **(f) Ours**

**Figure 10.** Comparison of dehazing image effects of different networks in real sea–skyline detection.

According to the experimental findings, the proposed maritime image dehazing network can improve the color spots and overall color dimming phenomenon existing in other networks to a certain extent. The recovered image is able to recover more details in the hazy region, which can provide higher-quality maritime images to the USV perception system. Table 5 also demonstrates that the sea–skyline accuracy of the image after processing by the proposed network and AOD-Net can both reach 95%, which is better than the other networks. Figure 10 shows that both the proposed network and AOD-Net outperformed the other networks in terms of the sharpness of sea–skyline part recovery, which is favorable

for sea–skyline identification, but the present network was more powerful in recovering the overall image quality. The sea–skyline detection on hazy days further validates the proposed network's excellence and efficiency applied to the USV perception system for maritime image dehazing.

**Table 5.** Comparison of detection accuracy of sea antenna after dehazing in different networks.

| Before/After Dehazing | Detection Success (Sheets) | Accuracy Rate (%) |
|:---:|:---:|:---:|
| Before dehazing | 12 | 60 |
| DCP | 12 | 60 |
| DehazeNet | 16 | 80 |
| AOD-Net | 19 | 95 |
| GCANet | 15 | 75 |
| **Ours** | 19 | 95 |

## 6. Running Time Comparison

The average dehazing time for each method is shown in Table 6. The test dataset for all methods was selected as 30 real-world images at the maritime surface; each method performed the dehazing operation on all images in the test dataset, and finally, the average of the dehazing time of all images was taken; moreover, the test work of all algorithms was performed on the same computer; meanwhile, all algorithms were implemented in the python programming language.

**Table 6.** Comparison of average model runtime.

| Methods | DCP | DehazeNet | AOD-Net | GCANet | Ours |
|:---:|:---:|:---:|:---:|:---:|:---:|
| Time (s) | 1.48 | 0.39 | 0.09 | 0.30 | 0.36 |

As can be seen from Table 6, the traditional DCP had the longest average time for dehazing, mainly because it is a staged algorithm for dehazing based on the atmospheric scattering model. Therefore, this method is much less efficient than the later proposed deep-learning-based end-to-end methods. In comparison to other deep-learning-based dehazing methods, AOD-Net had the highest dehazing efficiency. The main reason is that AOD-Net has a lightweight network structure, which makes the efficiency greatly improved. This unique feature also allows AOD-Net to be frequently used with high-level vision tasks such as target detection to improve the accuracy of target detection in hazy conditions. The dehazing efficiency of the method proposed in this paper was similar to the other two advanced methods, even higher than DehazeNet, the main reason being that the method uses a simple structure similar to U-Net and involves fewer modules. However, there is still much room for improvement compared to the processing speed of AOD-Net, and the proposed network needs to be further lightened.

## 7. Conclusions

In the paper, an end-to-end multi-branch gated fusion network was proposed for maritime image dehazing. To improve the defects of previous networks with a weak ability to retain high-frequency image information and poor detail recovery in thick haze regions, a residual attention structure was used to focus on important image features. Additionally, a gated fusion sub-network was employed to fuse various feature levels. Although the branch network is simple, it has great advantages over the previous state-of-the-art methods, especially in terms of color fidelity and thick haze region recovery. In the comparison with other networks, the advantages of the proposed network were demonstrated from both quantitative and qualitative perspectives. The practical application of sea–skyline detection

was also introduced in a maritime hazy condition, and it was further demonstrated that the proposed network can provide higher quality maritime images for USV vision system.

In the future, the dehazing time of the network will be further controlled, the network structure will be optimized, and the network will be lightened, so as to ultimately accomplish the goal of dehazing the USV perception system in real-time.

**Author Contributions:** Funding acquisition, Y.F.; writing—original draft, L.N.; writing—review and editing, T.L. All authors have read and agreed to the published version of the manuscript

**Funding:** This research was funded by the "National Natural Science Foundation of China" (Grant Numbers 61976033 and 51609033), the "Key Development Guidance Program of Liaoning Province of China" (Grant Number 2019JH8/10100100), the "Soft Science Research Program of Dalian City of China" (Grant Number 2019J11CY014), the "Fundamental Research Funds for the Central Universities" (Grant Numbers 3132021106 and 3132020110), and the "China Postdoctoral Science Foundation"(Grant number 2022M710569).

**Institutional Review Board Statement:** Not applicable.

**Informed Consent Statement:** Not applicable.

**Data Availability Statement:** Not applicable.

**Conflicts of Interest:** The authors declare no conflict of interest.

## Abbreviations

The following abbreviations are used in this manuscript:

| | |
|---|---|
| USV | unmanned surface vehicle |
| MGFNet | multi-branch gated fusion network |

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
