# Peer review of "Multi-Branch Gated Fusion Network: A Method That Provides Higher-Quality Images for the USV Perception System in Maritime Hazy Condition"

_jmse, doi:10.3390/jmse10121839_

Round 1

Reviewer 1 Report

"In order to focus on the trade-off between human perception and metric scores, new loss functions are proposed to optimize MGFNet, as shown in the following Eq. 11." If this is your proposal it should be emphasized. Namely, it is interesting scientific contribution. Even better if you prove that it is better in this application than other usual loss functions. In order to do so, you should add table with performance comparison to other usual loss functions (not just one, but all significant). I think that this part would be a greater contribution to the science than any other results presented in the paper. However, if this loss function is not your proposal, you should use it for comparison int he following section.

Check Table 2. PSNR should be smaller for better algorithms. If it is not the typical situation, you should explain it in the text that everyone sees it.

You should comment: https://doi.org/10.3390/rs13101950

https://arxiv.org/abs/2108.12863

Also you could comment if it fits the paper's final outlook:

http://dx.doi.org/10.1109/pic.2015.7489861

http://dx.doi.org/10.1049/el.2018.0989

https://doi.org/10.7225/toms.v09.n01.010

https://doi.org/10.1007/s00371-021-02321-0

You should explain the difference between horizon line and sea-sky-line detection. Is it the same algorithm useful for both? Is it the same or just similar? 

Reviewer 2 Report

Great paper, but I find a few remarks:

1. term CA Layer first meat in Tab.1 without explanation, please add it.

2. line 137: I can not distinguish 3 u-netes in FIg.1 , please clarify figure.

3. dash line in the right  part of Fig.3 is not clear.

4. literature review does not describe all state of the art in accordance with https://paperswithcode.com/task/image-dehazing. For instant, for RESIDE database the cite proposed LCANet as SOTA .

5.  I recommend you to add descriptions of  structural similarity and psnr calculation in your case, also please add cites if some authors in this area have already utilized them

6. section 4.3. Experimental dataset: please show examples of original and hazed images, if you have any ability also it would be great to add link on your database.

Round 2

Reviewer 1 Report

Paper always can be better by introducing more answers the reviewers' questions. However, it is improved from previous version.